# Electrophysical Properties of Polycrystalline C12A7:e⁻ Electride

**Alina A. Rybak** [1,2,*], **Ivan D. Yushkov** [1,3], **Nazar A. Nikolaev** [1,2], **Aleksandr V. Kapishnikov** [1,4],
**Alexander M. Volodin** [4], **Grigory K. Krivyakin** [1], **Gennadiy N. Kamaev** [3] and **Pavel V. Geydt** [1]

1   Laboratory of Functional Diagnostics of Low-Dimensional Structures for Nanoelectronics,
    Novosibirsk State University, Pirogova Str. 2, 630090 Novosibirsk, Russia; i.iushkov@g.nsu.ru (I.D.Y.);
    nazar@iae.nsk.su (N.A.N.); avl97@mail.ru (A.V.K.); grigory.krivyakin@gmail.com (G.K.K.);
    p.geydt@nsu.ru (P.V.G.)
2   Terahertz Photonics Group, Institute of Automation and Electrometry, Siberian Branch of the Russian
    Academy of Sciences, Koptug Ave. 1, 630090 Novosibirsk, Russia
3   Rzhanov Institute of Semiconductor Physics, Siberian Branch of the Russian Academy of Sciences,
    Lavrentyev Ave. 13, 630090 Novosibirsk, Russia; kamaev@isp.nsc.ru
4   Boreskov Institute of Catalysis, Siberian Branch of the Russian Academy of Sciences, Lavrentyev Ave. 5,
    630090 Novosibirsk, Russia; volodin@catalysis.ru
*   Correspondence: a.rybak1@g.nsu.ru

**Abstract:** This article demonstrates the possibility of creating memory devices based on polycrys-
talline mayenite. In the course of the study, structural characterization (XRD, TEM) of ceramic
samples of mayenite was carried out, as well as a study of the spectral (THz range) and electrophysi-
cal characteristics. Materials obtained by calcination at high (1360–1450 °C) temperatures in an inert
argon atmosphere differ in the degree of substitution of oxygen anions $O^{2-}$ for electrons, as indicated
by the data on the unit cell parameters and dielectric constant coefficients in the range of 0.2–1.3 THz,
as well as differences in the conducting properties of the samples under study by more than five
orders of magnitude, from the state of the dielectric for C12A7:$O^{2-}$ to the conducting (metal-like)
material in the state of the C12A7:e⁻ electride. Measurements of the current–voltage characteristics
of ceramic C12A7:e⁻ showed the presence of memristive states previously detected by other authors
only in the case of single crystals. The study of the stability of switching between states in terms of
resistance showed that the values of currents for states with high and low resistance remain constant
up to 180 switching cycles, which is two times higher than the known literature data on the stability
of similar prototypes of devices. It is shown that such samples can operate in a switch mode with
nonlinear resistance in the range of applied voltages from –1.3 to +1.3 V.

**Keywords:** memristor; polycrystalline electride; C12A7; I–V curves; material characterization

## 1. Introduction

The interest in materials based on calcium aluminates with a mayenite structure
($12CaO \cdot 7Al_2O_3$, designated as C12A7) is largely due to the possibility of varying their
functional properties over a wide range due to changes in the composition of the anionic
sublattice, which occur with the retention of the cationic framework of the mayenite
structure [1–6]. The positive charge of the $4^+$ cationic framework of $[Ca_{24}Al_{28}O_{64}]^{4+}$ is
compensated by the total negative charge of the $4^-$ anions included in the anion sublattice.
The chemical and electrophysical properties of such materials can be varied over a very
wide range by replacing the $X^-$ anions. The unit cell composition of such a compound can
be described by the Formula (1):

$$1 \text{ unit cell} = [Ca_{24}Al_{28}O_{64}]^{4+} \times 4X^-, \text{ where } X^- = H^-, O^-, O_2^-, O^{2-}, OH^-, Cl^-, F^-, e^- \quad (1)$$

The most interesting and unusual materials obtained based on mayenite are electrides
($X = e^-$) in which the role of anions is played by electrons. This class of materials was
first discovered in the works of the H. Hosono laboratory about 20 years ago [1,2] and it

was precisely these works that stimulated the development of work on the study of the properties of inorganic electrodes and the search for possible areas of their use. The greatest development in the practical use of these materials to date has received the direction of work associated with their use as electron-donor carriers for heterogeneous catalysts [7–9]. Along with this, several publications have demonstrated the possibility of using this class of inorganic electrodes as a material for creating components of electronic devices. The following are the main properties of C12A7 material in terms of its possible use as a material for the creation of components of electronic devices:

1.  The low work function (2.1–2.4 eV) makes it possible in principle to use the C12A7:e$^-$ electrode for creating economical and efficient cathodes in a variety of devices. Including those using the thermal emission properties of the material [10–12]. Recently, the possible use of this material in field-effect transistors, light-emitting diodes, and also as emission elements in displays [13,14] has been discussed. In the reviews [15,16], the possible areas of using this material for the creation of components of electronic devices are presented in sufficient detail.

2.  Another possibility of using the C12A7 material is the creation of memristive memory elements based on it. The principle of operation of such elements is based on the reversible diffusion of oxygen ions in the C12A7:e$^-$ /C12A7:O$^{2-}$—a system under the action of an electric field. The fundamental possibility of creating such elements was demonstrated in [17] and is also reflected in the review [16]. It should be noted that a single-crystal material was used in this work as a material for creating such elements. At the same time, of considerable interest is the practical use of polycrystalline materials, the technology of obtaining and controlling the conductivity (from the state of the dielectric to metallic) which can be much simpler and cheaper.

This work aimed to obtain information on the electrophysical and spectral characteristics of several polycrystalline samples of mayenite with various degrees of substitution of oxygen anions by electrons and to assess the possibility of the formation of memristive states in such systems.

## 2. Materials and Methods

### 2.1. Fabrication Technique and Fabrication Setup

In this work, the method of synthesis of mayenite previously described in [18–20] was used. Aluminum hydroxide (Condea Pural SB—1, pseudo-boehmite, >99.9%) and calcium carbonate (Reachim, >99.97%) were used as starting materials for the synthesis. The first stage of the synthesis was to obtain CaO by decomposing $CaCO_3$ in a muffle furnace in the air at 950 °C for 6 h. After that, the resulting CaO was added to a suspension of aluminum hydroxide in distilled water at room temperature with continuous stirring. The final ratio of the components corresponds to the stoichiometry of mayenite ($12CaO \cdot 7Al_2O_3$). The mixture was thoroughly stirred in distilled water for 10 h, filtered, and dried at 110 °C. Then, it was calcined in a muffle furnace in air at 500 °C for 6 h. The samples obtained were successively calcined in an Ar flow at high temperatures (>1300 °C). The initial sample C12A7-500 was heated at a rate of 3 °C/min, followed by holding at the specified temperature for 6 h. For characterization and investigation of electro-physical properties, a sample was taken at four temperature points: at 1360 °C, 1380 °C, 1390 °C, and 1450 °C. The appearance of the selected samples is shown in Figure 1, which in the future will be designated, respectively, as C12A7-1360 (white sample), C12A7-1380 (green), C12A7-1390 (dark green), and C12A7-1450 (black, electride).

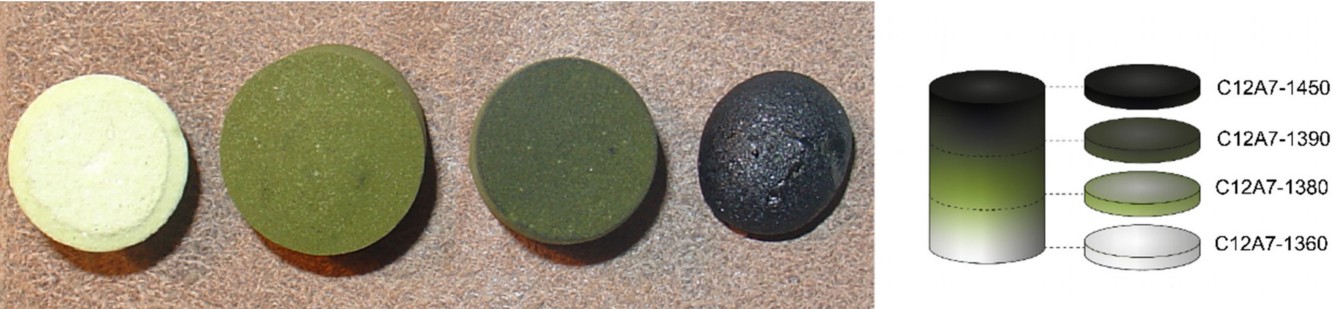

**Figure 1.** Photography and cutting scheme of samples C12A7-1360, C12A7-1380, C12A7-1390, and C12A7-1450 (from **left** to **right**).

## 2.2. Characterization Methods

X-ray powder diffraction studies were performed on an ARL X'tra diffractometer (ThermoFisher Scientific, Switzerland, Basel) with a radiation source Cu–Ka ($\lambda$ = 1.5418 Å). The recording of diffraction patterns was carried out in the range of $2\theta$ = 15–85°. The figures show inserts in the range up to 60°. The step was 0.05°, the signal accumulation time was 3 s at each point. The processing of diffraction patterns and semiquantitative phase analysis was carried out by refining the full profile by the Rietveld method in the GSAS-II program [21].

HRTEM images were obtained on a JEM-2200FS transmission electron microscope (JEOL, Japan) with a spherical aberration corrector at an accelerating voltage of 200 kV. Initially, a sample in the form of a finely dispersed powder was deposited on the grid using an ultrasonic disperser.

To measure absorption spectra in the THz range, disks about 1 mm thick were cut out of the samples. The measurements were carried out using a small-sized broadband pulsed terahertz time–domain spectrometer (THz–TDS) located at the Institute of Automation and Electrometry SB RAS, the spectral range of which is 0.1–2.7 THz, with a peak in the spectral dynamic range of >70 dB at a frequency of 0.3 THz. A detailed description of the setup is given in [22,23]. The schematic of the THz–TDS is shown in Figure 2.

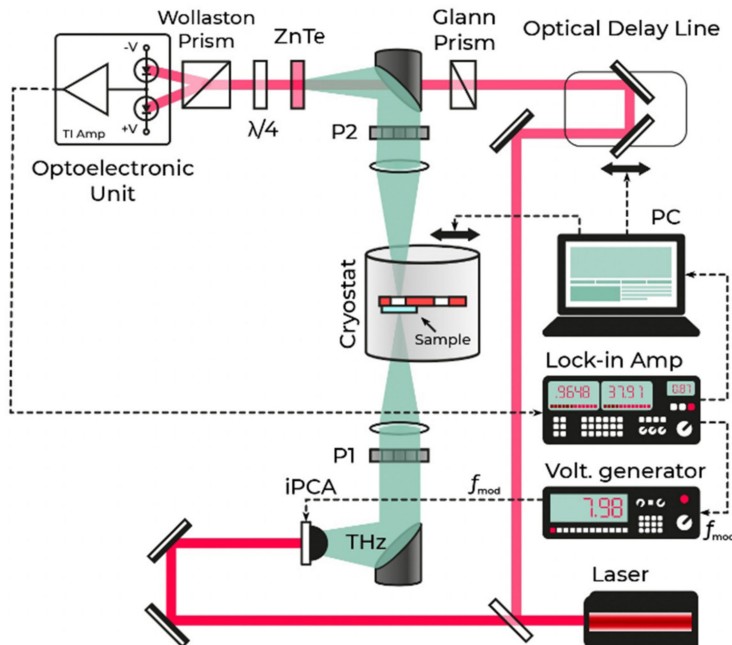

**Figure 2.** Schematic diagram of the THz–TDS.

The scanning range of the THz–TDS is 60 ps, which corresponds to a spectral resolution of 20 GHz. The scanning step was 125 fs. The averaging time at each point was 0.21 s with the time constant of the lock-in amplifier filter equal to 30 ms. In the study, a liquid nitrogen cryostat with fused silica windows and a resistive heater was used. The sample holder had two diaphragms with a diameter of 7 mm in the lower flat part (Figure 2). The sample under study was fixed on one of the diaphragms, the other was used to record reference THz signals. After processing terahertz signals in the same way as described in [24], the values of the real and imaginary parts of the permittivity were obtained.

Porosity of the polycrystalline samples can be the main factor that influences permittivity so we made additional recalculations. The density values were measured by the helium pycnometry method on the automatic ULTRAPYC-1200e device (Quantachrome Instruments, Boynton Beach, FL, USA) with $\pm 0.03\%$ accuracy. Obtained values were used for the estimation of sample porosity.

Influence of porosity was taken into account using the Maxwell–Garnett effective medium model with spherical inclusions [25]:

$$\widetilde{\varepsilon}_{eff} = \widetilde{\varepsilon}_m \frac{(\widetilde{\varepsilon}_a + 2\widetilde{\varepsilon}_m) + 2\eta(\widetilde{\varepsilon}_a - \widetilde{\varepsilon}_m)}{(\widetilde{\varepsilon}_a + 2\widetilde{\varepsilon}_m) - \eta(\widetilde{\varepsilon}_a - \widetilde{\varepsilon}_m)} \tag{2}$$

where $\widetilde{\varepsilon}_{eff}$—is the initially measured complex dielectric constant of the samples, $\widetilde{\varepsilon}_a$—dielectric constant of inclusion (air), $\widetilde{\varepsilon}_m$—the required dielectric constant of the medium (medium) i.e., mayenite, is the factor of filling the medium with particles (the ratio of the total volume of voids to the entire volume of the medium).

The calculations were carried out in such a way that $\mathrm{Re}[\widetilde{\varepsilon}_m]$ of the mayenite sample coincided with the value of $\varepsilon_1$ for the single-crystal mayenite samples presented in [26,27] since the material density makes the main contribution to the real part of the dielectric susceptibility.

The study of the electrophysical characteristics of the obtained samples was carried out on metal–mayenite–metal structures. The upper nickel electrodes with a thickness of 200 nm and an area of 0.5 mm$^2$ were deposited through a mask by magnetron sputtering in an Ar atmosphere. On the reverse side, a continuous nickel layer of the same thickness was deposited. The I–V characteristics (current–voltage characteristics) of the structures obtained and the stability of the operation of the C12A7-1450 sample were measured at room temperature in DC mode using a B2902A two-channel source-meter (Agilent, Santa Clara, CA, USA) with micropositioners with probes connected to it.

The voltage sweep was performed from 0 to 50 V for samples C12A7-1360, C12A7-1380, and C12A7-1390, and from –2 to 2 V for C12A7-1450. The measurements were carried out stepwise at the end of each step. The measurement duration at each point was 0.2 s for C12A7-1360, C12A7-1380, and C12A7-1390, and 0.5 s for C12A7-1450. The measurement parameters were set and the data were read out using the Agilent B2900A Quick I/V Measurement Software.

## 3. Results and Discussion

### 3.1. XRD Analysis

The samples were analyzed by XRD to confirm their phase composition and structure of the obtained material. According to XRD data (Figure 3, Table 1), the mayenite sample C12A7-1360 predominantly contains the $Ca_{12}Al_{14}O_{33}$ phase (C12A7, JCPDS No 9-413), with small impurities of $Ca_3Al_2O_6$ (C3A, JCPDS No 38-1429) and $CaAl_2O_4$ (CA, JCPDS No 23-1036). In the sample C12A7-1380, in that the formation of the electride state C12A7:e$^-$ began, in addition to the main phase, a large amount (from 20 to 40%) of impurity phases C3A and CA were detected. It should be noted that both C12A7-1380 and C12A7-1390 samples demonstrated similar diffractograms. At the same time, C12A7-1450 is a single-phase material (up to 95%) with an insignificant admixture of CA.

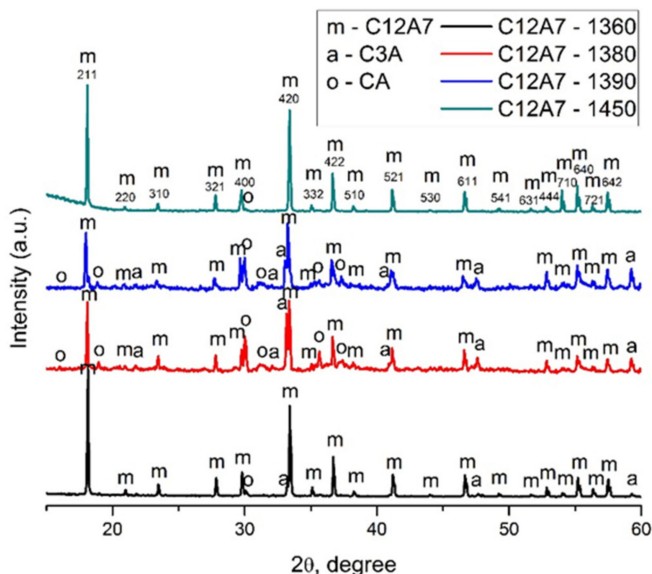

**Figure 3.** XRD patterns of the C12A7-samples upon calcination from 1360 to 1450 °C.

**Table 1.** XRD parameters for C12A7 samples.

| T, °C | Phase Composition | a, Å |
|---|---|---|
| 1360 | $Ca_{12}Al_{14}O_{33}$ (89%) + $Ca_3Al_2O_6$ (6%) + $CaAl_2O_4$ (5%) | 11.998 ± 0.004 |
| 1380 | $Ca_{12}Al_{14}O_{33}$ (44%) + $Ca_3Al_2O_6$ (20%) + $CaAl_2O_4$ (36%) | 12.001 ± 0.003 |
| 1390 | $Ca_{12}Al_{14}O_{33}$ (55%) + $Ca_3Al_2O_6$ (14%) + $CaAl_2O_4$ (31%) | 12.010 ± 0.007 |
| 1450 | $Ca_{12}Al_{14}O_{33}$ (95%) + $CaAl_2O_4$ (5%) | 12.004 ± 0.002 |

The lattice parameter of the mayenite phase (Table 1) slightly increases during these transformations from 11.998 to 12.004 Å. A gradual increase in the calcination temperature leads to the decomposition of the mayenite phase into phases C3A and CA, the content of which in the system is rather high. In the literature [28,29], it was indicated earlier that the process of oxygen release from the structure can be accompanied by rearrangements of the latter, with its stratification into aluminates of a different composition. Then, with a gradual increase in temperature in an inert atmosphere, an oxygen-deficient electride state $C12A7:e^-$ is formed. The increased value of the lattice parameter indicates the presence of oxygen vacancies at the crystallographic O3 positions, and, as a consequence, the presence of electrons in the unit cells of mayenite [30,31].

### 3.2. TEM Analysis

The HRTEM micrographs show the microstructure of the initial C12A7-1360 (Figure 4a,b) and final C12A7-1450 (Figure 4c,d) samples. The morphology of samples was presented by flat well-crystallized particles of large size (0.5–5 μm). It is clearly shown that samples have the analogous microstructure and no significant differences in it. To estimate the interplanar distances of the phase, a Fourier transform of the obtained images of the crystal structure was carried out. The obtained values, in particular 0.4898 nm, are in good agreement with the XRD data (0.4897 nm for the 211 reflection). This also confirms the presence of a mayenite-type structure.

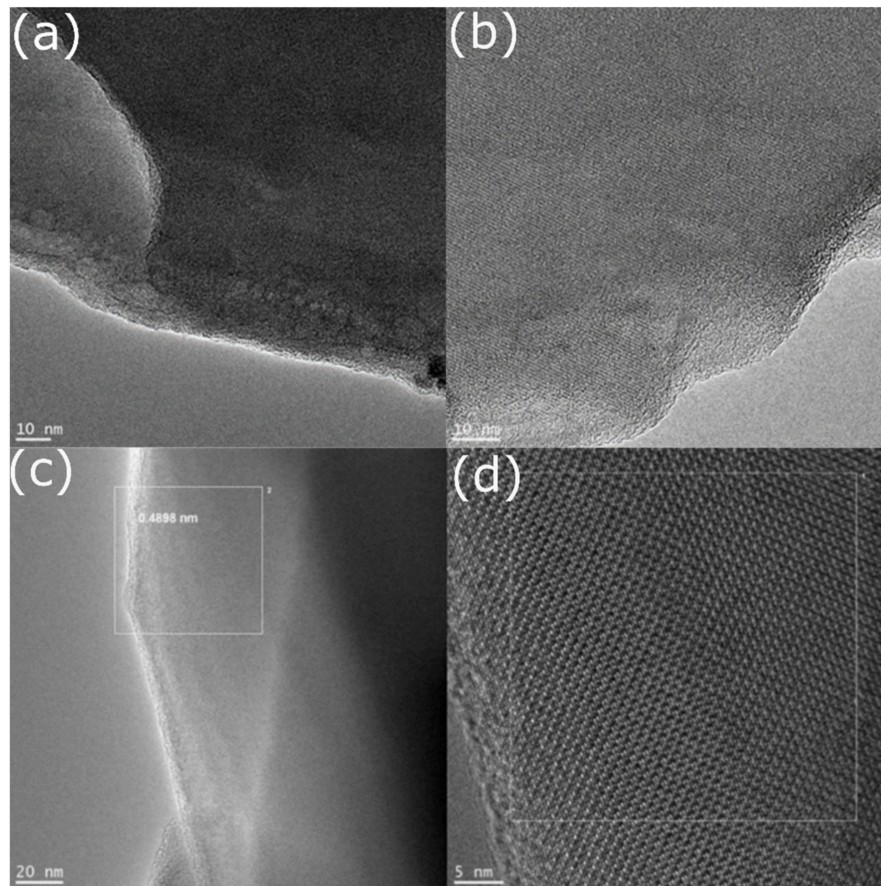

**Figure 4.** Microstructure of the mayenite particles in samples C12A7-1360 (**a**,**b**) and C12A7-1450 (**c**,**d**).

### 3.3. Terahertz Spectroscopy

The value of the measured density is approximately 2.65 g/cm$^3$. Thus, we specified that $\eta$ = 0.13 (Equation (2)), which corresponds to a density of 87% for our initially synthesized samples. The resulting values of the real $\varepsilon_1$ and imaginary $\varepsilon_2$ parts of the dielectric constant of our materials are shown in Figure 5. For comparison, we also adapted the data from the article [26] (C12A7-P, C12A7-H), with which there is a good agreement.

An analysis of the spectra shows that ongoing from pure mayenite to the electric phase, an increase in $\varepsilon_2$ to low frequencies is observed. This is in good agreement with the data obtained by Hosono et al. (see the curve for C12A7-H in Figure 5) and indicates an increase in the concentration of free charge carriers, which is expressed in an increase in the plasma frequency and a gradual increase in attenuation for the low-frequency part of the spectrum. We also note that this is also the reason why we were unable to study the C12A7-1450 electride sample. Since the concentration of free charge carriers in it is so high that the plasma frequency already lies above the spectral range of our THz–TDS, this leads to total reflection of the THz wave from the sample and does not allow obtaining a signal in transmission geometry. From the $\varepsilon_2$ graphs in Figure 5, we can conclude that the concentration of charge carriers in the C12A7-1390 sample corresponds to the values for C12A7-H from [26] and has a value of ~$10^{18}$ cm$^{-3}$. At the same time for C12A7-1380, this value is noticeably lower. A sharper increase in $\varepsilon_2$ in our case is most likely associated with the scattering of high-frequency THz radiation by air pores in the samples.

Cooling of our samples did not significantly affect $\varepsilon_1$. However, a noticeable change was observed in $\varepsilon_2$, in particular, a decrease in the low-frequency region. Presumably, this is due to the capture of free charge carriers during cooling of the material.

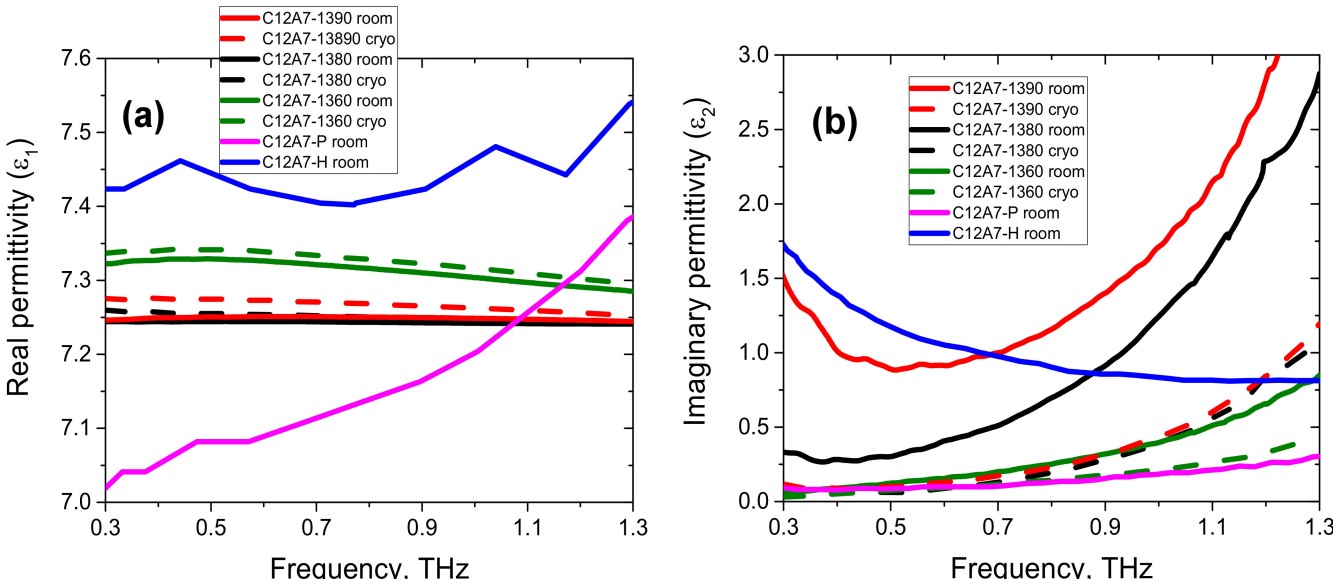

**Figure 5.** Spectra of the real (**a**) and imaginary (**b**) parts of the dielectric constant of mayenite, measured at temperatures of 24 °C (room) and −192 °C (cryo).

### 3.4. Electrophysical Characterization (I–V Curves)

The I–V characteristics of all samples are shown in Figure 6. Three samples have shown hysteresis, which is characteristic of dielectrics. The currents for samples C12A7-1360, C12A7-1380, and C12A7-1390 are $3.2 \times 10^{-11}$ A, $2.5 \times 10^{-8}$ A, and $6.2 \times 10^{-9}$ A, respectively, at a voltage of 200 V. High voltages were not applied to the sample C12A7-1450 to preserve the integrity of the sample, although, an almost vertical I–V curve was observed. This indicates an exceedingly lower resistance of C12A7-1450 compared with other samples.

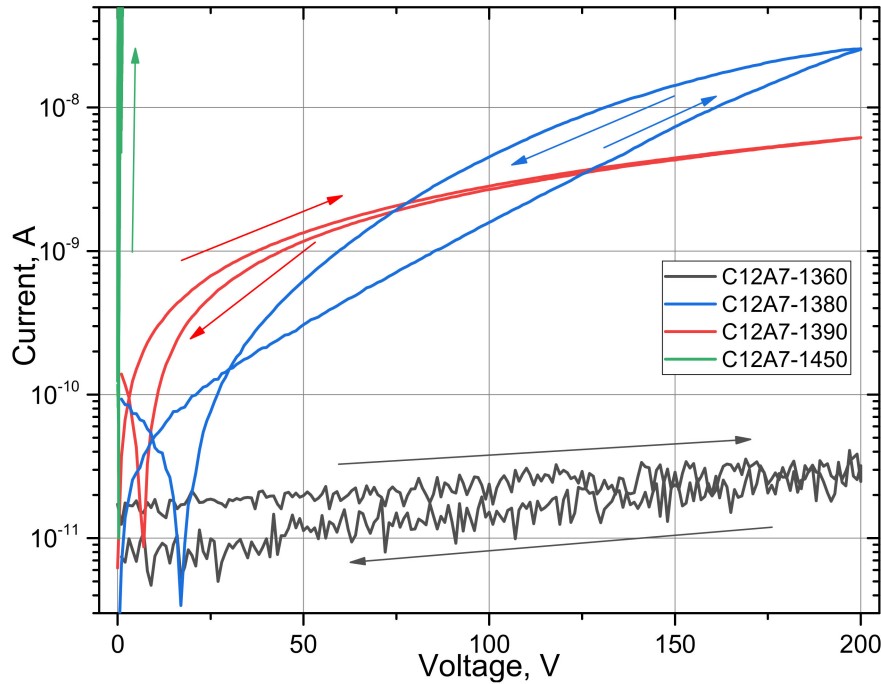

**Figure 6.** I–V characteristics of samples C12A7-1360, C12A7-1380, C12A7-1390, and C12A7-1450. The arrows indicate the direction of the recording.

The difference in current between samples C12A7-1360, C12A7-1380, and C12A7-1390 can be explained by the fact that at a temperature of 1380 °C oxygen ions $O^{2-}$ are replaced by electrons $e^-$, and an electride phase is formed. In connection with this, the concentration of free charge carriers (presumably electrons) increases. This suggestion is also confirmed by the spectra analysis of the imaginary part of the dielectric constant $\varepsilon_2$.

The I–V characteristic of the C12A7-1380 sample increases exponentially, which indicates the diode properties of the sample. In addition, the higher current during scanning in the opposite direction is explained by the low response rate of the internal electric field of the C12A7-1380 sample. Most likely, this behavior is associated with the uneven formation of the electric mayenite phase over the thickness during heat treatment.

The first I–V characteristics of the C12A7-1450 sample were studied in the range of –2 to 2 V with a voltage step of 0.01 V and the current limitation of $10^{-6}$ A (Figure 7). The arrows show the direction of the data scanning. The measurements started at 0 V and continued in the direction of negative voltages. It can be seen from the I–V characteristic that hysteresis exists at negative polarity. When the voltage reaches −1.4 V, a sharp current increase occurs, and the source switches to the current mode due to the limitation of the current value (in our case, $10^{-6}$ A). During the reverse stroke, the structure is at this current limitation down to −1 V and then the sample exhibits a lower resistance. That is, at −1.4 V, there was a transition from a high-resistance (HR) state to a low-resistance (LR) state. The window between states in this region is two orders of magnitude in current. Then, a positive bias from 0 to 1.6 V was applied to the structure. With increasing voltage, the current increases and reaches the specified limit. Furthermore, with decreasing voltage, the current also decreases already in a state with lower resistance. In the region of positive voltages, the I–V characteristics are unstable, which indicates unipolar switching between the states, i.e., switching occurs only in one polarity. Similar I–V characteristics are observed in memristor structures [32,33]. Moreover, in the second half of the cycle, there is a zero-current bias in the HR state. This feature indicates the accumulation of charges in the bulk of the structure [27]. The switching effects from high-resistance to low-resistance states and back, typical of memristors, were not observed for the samples C12A7-1360, C12A7-1380, and C12A7-1390. Therefore, the memristor properties of sample C12A7-1450 are described in detail below.

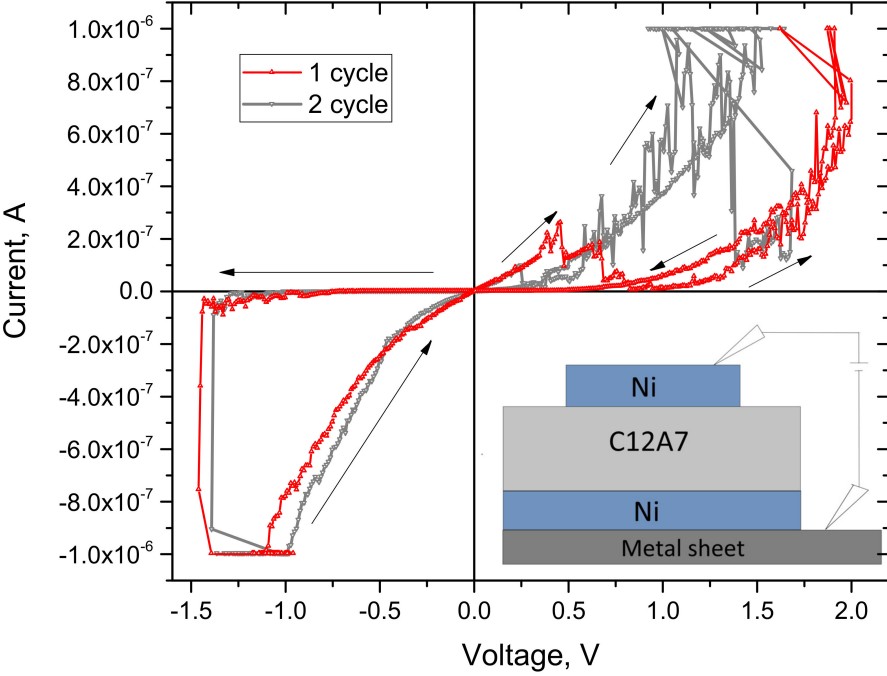

**Figure 7.** I–V characteristic of the C12A7-1450 sample. The arrows indicate the direction of the recording.

Figure 8 shows the I–V characteristics for the C12A7-1450 sample with three sweep cycles. The variable parameter in this group of measurements was the initial scanning voltage of 0.6 V, 1 V, and 1.6 V. At an initial point of 0.6 V, the switching voltage was –0.68 V, while at 1 V, the switching voltage was −0.92 V, and when the initial point was increased to 1.6 V, the switching voltage became −1.16 V (in Figure 8, the switching points are denoted as *A*, *B*, and *C*, respectively). Arrows indicate the direction of data measurement. Thus, Figure 8 illustrates the dependence of the switching voltage on the starting point. The switching voltage between the states (in terms of resistance) increases with an increase in the initial voltage.

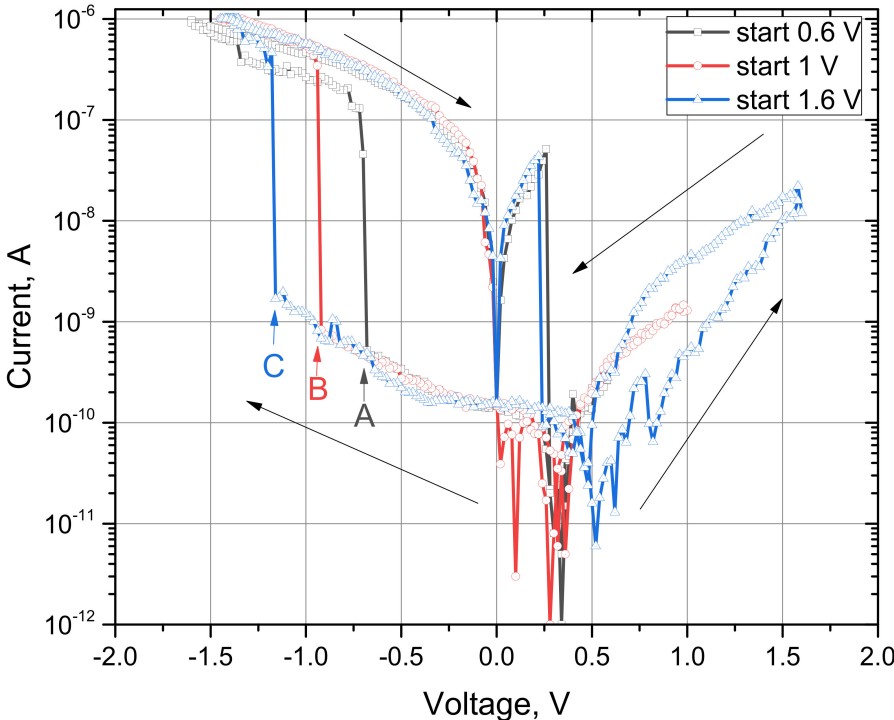

**Figure 8.** I–V characteristics of the C12A7-1450 sample at different initial voltages.

A study of the LR-state on-state current stability is represented in Figure 9. The measurements were carried out stepwise according to the scheme shown in Figure 9b. The current limit was $10^{-6}$ A. In the first few seconds (Figure 9a), there is a switch between off and on states. This sets the current corresponding to this voltage. Then, after 50 s, the voltage decreases in steps to −1.05 V and it is maintained at this value for the next 50 s. The current corresponding to this voltage also decreases and remains stable until the next voltage change of −0.5V. The current in the sections with a voltage of −1.05 V and −0.5 V is stable and corresponds to the current in the I–V characteristic in Figures 7 and 8. Furthermore, the voltage also rises in steps to the initial value of −1.6V. As the voltage changes, the current also increases. Thus, this sample can operate in the mode of a switch with a nonlinear resistance.

Furthermore, on-state storage studies were conducted in order to verify the memristor characteristic of C12A7-1450. Initially, the sample was switched to an LR state. After switching to a different state, the contacts on the sample were disconnected from the power supply for a few minutes (<10 min). Furthermore, the current dependence on time was measured (Figure 10) at a constant voltage of –0.5 V. When disconnected from the network current it was about $5 \times 10^{-8}$ A, which is much less than it should be on the I–V characteristic in Figure 9. The current decreased from $10^{-7}$ to $10^{-8}$ in a few minutes without a power supply. A decrease in the current indicates a very short storage time for the LR state and volatility, i.e., energy dependence, of the C12A7-1450 sample.

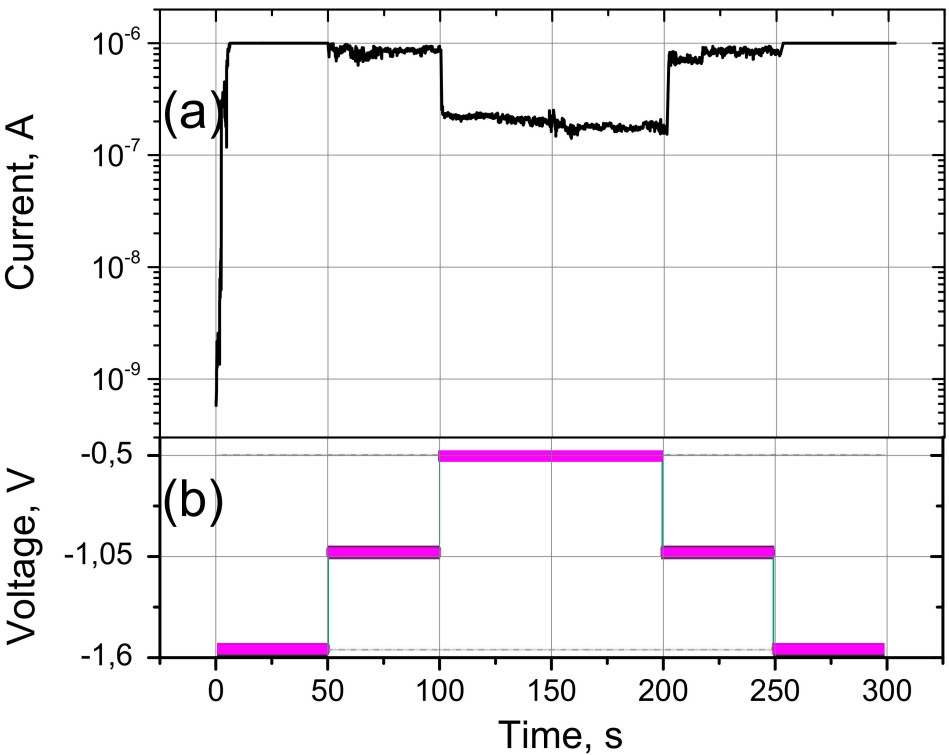

**Figure 9.** Investigation of current stability: (**a**) current versus time for the sample C12A7-1450 and (**b**) measurement scheme.

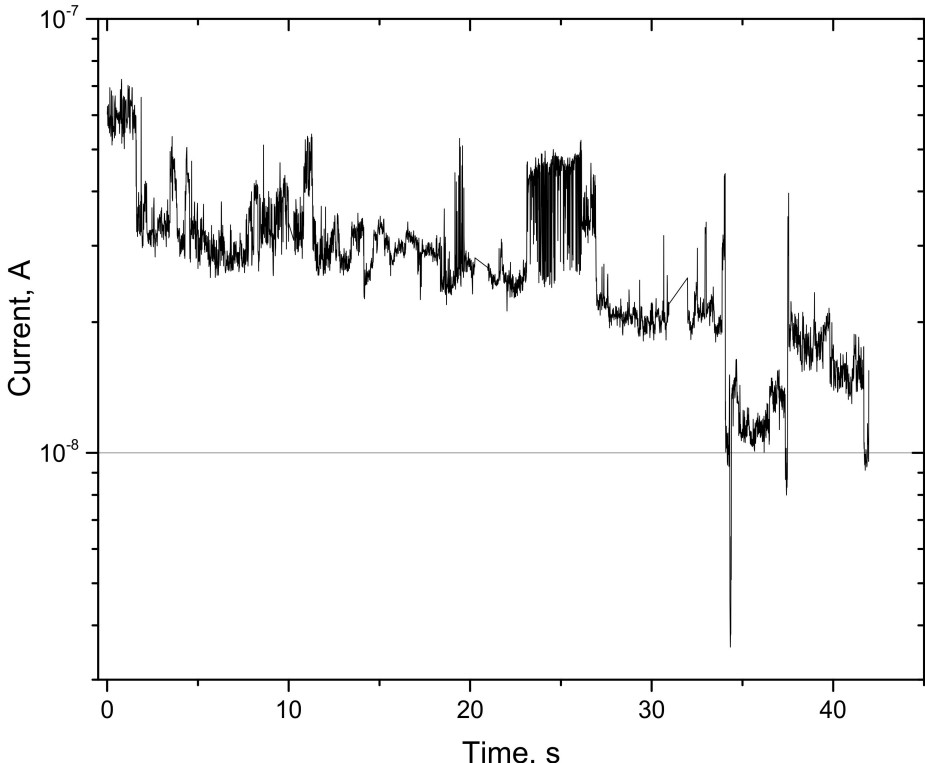

**Figure 10.** Dependence of current on time after disconnection from the network for the sample C12A7-1450.

Figure 11 shows a study of the stability of switching between the states in terms of resistance. One switching cycle begins with a voltage pulse corresponding to the switch–off voltage. After that, at a given readout voltage, the current was measured, which corresponds to the off stage. Then, a voltage pulse was applied corresponding to the turn-on voltage. After that, at a given readout voltage, the current was measured, which corresponds to the turn-on stage. Furthermore, the switching cycles are repeated N times. The duration of the switching pulses was 100 ms, with a period of 500 ms. The amplitude of the two corresponding pulses was 1 and –1.6 V, respectively. The reading voltage was –0.5 V. In Figure 11, it is noticeable that the switching between states was relatively stable. However, one can notice several cycles that have not switched to a LR state. The resistances were in the order of $10^9$ $\Omega$ in the off state and $5 \times 10^6$ $\Omega$ in the on state. In the other work on the study of bistable resistive switching of C12A7, cyclic switching was also investigated [17]. By comparing the data, it can be seen that the number of withstood cycles is higher [17]. Comparable dependencies were reported for other memristor materials [32].

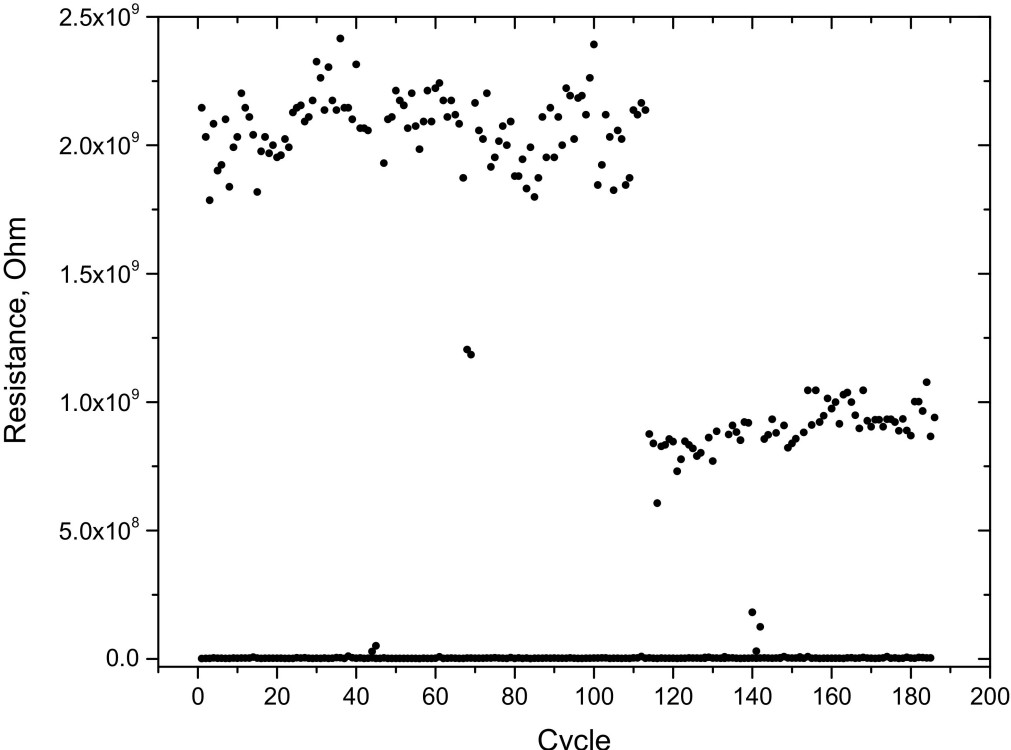

**Figure 11.** Cyclic switching between states by resistance for the sample C12A7-1450.

## 4. Conclusions

The possibility of creating a memory element based on polycrystalline mayenite synthesized from hydroxide precursors has been demonstrated. Materials after heat treatment at high (1360–1450 °C) temperatures in an inert argon atmosphere differ in the degree of substitution of electrons with oxygen anions $O^{2-}$ as evidenced by the data on the unit cell parameters and dielectric constant in the range 0.2–1.3 THz. There is a noticeable difference in the conductive properties of the samples in this study. Measurements of the I–V characteristics of the polycrystalline C12A7:$e^-$ ceramics showed the presence of memristive states, previously discovered by other researchers only in single-crystal materials. The study of the stability of switching between the conductive states in terms of resistance showed that the values of current for states with high and low resistance remain constant up to 180 switching cycles, which is two times higher than the known literature data on the stability of similar prototypes of memory devices. Such samples can operate in a switching mode with a nonlinear resistance in the range of applied voltages from –1.3 to

+1.3 V. It paves a way for the production of switch elements based on polycrystalline C12A7 materials. Such elements are analogous memristors that can be used as an elemental base for creation of analog neural networks.

**Author Contributions:** Conceptualization, N.A.N. and A.M.V.; data curation, I.D.Y.; funding acquisition, P.V.G.; investigation, A.A.R., I.D.Y., N.A.N., A.V.K., G.N.K. and G.K.K.; methodology, A.V.K., G.N.K. and A.M.V.; project administration, P.V.G.; resources, A.M.V.; visualization, A.A.R.; writing—original draft, A.A.R., I.D.Y., A.V.K. and P.V.G.; writing—review and editing, A.A.R. All authors have read and agreed to the published version of the manuscript.

**Funding:** This research was funded by the Ministry of Education and Science of the Russian Federation, grant No. FSUS-2020-0029, in terms of sample preparation and analysis.

**Acknowledgments:** The authors acknowledge the NSU Shared Equipment Centers "Applied Physics" and "VTAN". The authors acknowledge the Shared Equipment Center "Spectroscopy and Optics" of the Institute of Automation and Electrometry SB RAS for the provided terahertz time-domain spectrometer. The authors acknowledge M. I. Mironova for help with electron microscopy, V. A. Volodin for help with Raman spectroscopy, A. O. Geydt for help with translation, and An. A. Rybak for help with drawings.

**Conflicts of Interest:** The authors declare no conflict of interest. The funders had no role in the design of the study; in the collection, analyses, or interpretation of data; in the writing of the manuscript, or in the decision to publish the results.

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
