# Peer review of "Electrophysical Properties of Polycrystalline C12A7:e Electride"

_electronics, doi:10.3390/electronics11040668_

Round 1

Reviewer 1 Report

The goal of the work is to obtain information on the electrophysical and spectral characteristics of several polycrystalline samples of mayenite which may be potentially used in memory device. The topic is interesting because memory devices have attracted considerable research interests, but a large number of additional results have to be collected and shown to improve this manuscript. In addition, some figures, which are not very important to show in the main text, can be moved to Supporting Information to make the manuscript shorter and clearer. 

(1) The writing has to be improved because there are a number of grammar errors, improper expressions, and typos.

(2) It is better to explain the methods instead of using reference number only, although the methods have been reported.

(3) In the manuscript, authors have tried 4 different treatment conditions. Why did authors only show three samples in the photograph of Fig. 1? In addition, please explain why the samples shown in the photography have different sizes.

(4) Authors noted that both C12A7-1380 and C12A7-1390 samples demonstrated indistinguishable diffractograms. However, it is better to show XRD data of C12A7-1390 in Fig. 3 as well because this can be helpful for readers to compare them clearly.

(5) Authors studied materials after heat treatment at high temperature from 1360-1450 degrees, but they only show the data and results of the treatment under 1450 degrees in Fig. 4, Fig. 9, Fig. 10, and Fig. 11. In order to prove authors did the same measurements for other conditions, it is better to show the data and results in Supporting Information. 

(6) Please label (a) and (b) in Fig. 5 and clarify what cryo in Fig. 5 means in the main text.

(7) Please add several sentences to explain the significance of Fig. 9 and what it can indicate. 

Reviewer 2 Report

Authors have synthesized polycrystalline mayenite and annealed at high temperature in inert ambient. Finally tested them as resistive memory application. The manuscript is well written and properly explained, except few minor mistakes. Before publishing please make correction of these minor issues.

  1. Figure 1 caption mentioning 4 samples but actual optical picture showing only three of them please correct it.
  2. Authors showed XRD patterns of C12A7 (1360-1450). How about 500oC annealed sample, as well as as prepared or synthesized one?
  3. Figure 4 showing TEM image of C12A7-1450 only. Can you add other samples? or describe them?
  4. Page 6 line 196 "(see the curve for C12A7-H)" but which figure?

Round 2

Reviewer 1 Report

The revised manuscript is much better than the original manuscript after adding more helpful results and explanations. In addition, the structure of the manuscript is clearer and easier for readers to understand. However, there is a small question in Fig. 4 (SEM). It is better to add (a) to (d) in the figures and explain which figures show the microstructure of samples C12A7-1360 and C12A7-1450, respectively. Otherwise, readers cannot get information from the figure directly without reading the main texts. 

Author Response

Thank you for attention. We have corrected the Fig. 4 according to your comments.